# Disclosing Gender-Based Violence: A Qualitative Analysis of Professionals’ and Women’s Perspectives through a Discursive Approach

**DOI:** 10.3390/ijerph192214683

**Published:** 2022-11-09

**Authors:** Isabel Goicolea, Carmen Vives-Cases, Esther Castellanos-Torres, Erica Briones-Vozmediano, Belén Sanz-Barbero

**Affiliations:** 1Department of Epidemiology and Global Health, Umeå University, 90187 Umea, Sweden; 2Department of Community Nursing, Preventive Medicine and Public Health and the History of Science, University of Alicante, 03009 Alicante, Spain; 3CIBER of Epidemiology and Public Health (CIBERESP), 28029 Madrid, Spain; 4Public Health Research Group, University of Alicante, 03009 Alicante, Spain; 5Group of Studies in Society, Education and Health (GESEC), Nursing and Physiotherapy Department, University of Lleida, 25008 Lleida, Spain; 6Healthcare Research Group (GRECS), Biomedical Research Institute of Lleida (IRB), Josep Pifarré Fundation, 25198 Lleida, Spain; 7Department of Epidemiology and Biostatistics, Institute of Health Carlos III, 28029 Madrid, Spain

**Keywords:** gender-based violence, disclosure, reflexive thematic analysis, Spain, discourse

## Abstract

Supporting women to disclose gender-based violence (GBV) is a central feature of how healthcare and other welfare services address this problem. In this paper we take a discursive approach to analyse the process of disclosing GBV from the perspectives of young women who have been subjected to GBV and professionals working in the welfare system. Through a reflective thematic analysis of 13 interviews with young women who have been subjected to GBV and 17 with professionals working in different sectors of the welfare system, we developed four themes about how disclosure is perceived: (i) as a conversation between acquaintances; (ii) as ‘no solution’; (iii) as a possible prerequisite for action; and (iv) as difficult because GBV is normalised. Even if disclosure is not the solution per se, it makes it possible to respond institutionally to GBV on an individual basis through the figure of the expert professional who is alert to signs, knows how to support disclosure, and has the power to legitimate women’s claims of GBV. We acknowledge the possibilities that supporting disclosure brings for women subjected to GBV, but at the same time, problematise that it can re-centre expertise in the professional and place the responsibility on women.

## 1. Introduction


*There is a strong concern in relation to the low detection of gender-based violence […] In order to address this, it is a priority that healthcare services detect early gender-based violence.*
(Spanish Common Protocol for the Health System Response to Gender-Based Violence, [1] (p. 41))

The Spanish protocol for the health system response to gender-based violence (GBV), which we quote above, highlights low detection as a crucial gap in addressing GBV [1]. Later on, the protocol elaborates upon how detection can be facilitated both through being attentive to signs and symptoms and, especially, through encouraging women to disclose GBV by asking them about it. Disclosing GBV is presented as a crucial step, and protocols and models that focus on facilitating disclosure by women using public services (not only healthcare but also social services, education, and other factors) have become mainstream, both in Spain and internationally [1,2,3].

In most countries around the world, GBV has shifted from being mainly addressed by feminist civil society organisations to becoming recognised as a social problem for which the welfare system has a major responsibility [4]. In Spain, the 2004 Law on Protective Measures against Gender-based Violence details the responsibilities of different sectors, including social services, educational and legal systems, and healthcare services. Detecting violence early through encouraging women to disclose their experiences is part of what the welfare system should do.

The way in which public welfare services address GBV centres on professionals supporting women to realise that they have a problem and to take steps towards changing the situation. The rationale seems to be straightforward: in order to receive support, women need to become aware that they have a problem and tell it to someone who can provide such support [5,6,7,8]. The underlying assumption also seems straightforward: if professionals become aware that the woman they are meeting has been subjected to GBV (because she discloses it to them), they are able to offer her support directly or through referring her to appropriate resources. Such assumptions conceptualise disclosure as a crucial step that opens up the possibility of providing support.

In this paper, we want to scrutinise such assumption by analysing the process of disclosing GBV from the perspective of young women who have been subjected to GBV and professionals working in different services of the welfare system in the region of Madrid, Spain (healthcare, social services, education).

### 1.1. Disclosure of GBV

Traditionally, disclosure has been conceptualised as the revealing of one’s deepest thoughts and feelings surrounding upsetting events through speech or writing [9]. The concept has been used in the field of HIV, and to a certain extent, in relation to GBV and other forms of violence victimisation, such as sexual abuse and dating violence [10,11,12].

Research depicts disclosure of GBV as beneficial, while inhibiting disclosure is considered to be a stressor [9,13]. Rhodes et al., for example, state that women who disclosed GBV to healthcare practitioners experienced it as ‘life-changing’. However, they also acknowledged that this beneficial effect was dependent on the disclosure resulting in non-judgmental suggestions [14]. Disclosing is not (always) easy since it involves uncertainty and the risk of rejection and judgement [15]. When it comes to disclosing GBV, the literature reveals that the most common person to disclose to is a friend or relative and not a professional [16].

Research has also described factors that facilitate disclosure, e.g., clearly defining the terms and conditions, listening and responding with genuine emotion, showing interest and ensuring that it is done in an open, unobtrusive way [15,17]. Thus, the benefits of disclosure are subject to *how* it takes place [15,18], with research showing that negative experiences—such as reactions of denial—may discourage future disclosure [16]. Research has also determined several factors that hinder disclosing GBV to professionals, such as fear about the perpetrators’ reaction, lack of trust in professionals’ responses, guilt and shame, and lack of information about support services and rights as well as the challenges that come with recognising GBV and adopting a ‘victim/survivor’ position [19,20].

### 1.2. Our Conceptual Approach—A Discursive Approach to Disclosure

Traditional models of disclosure portray the one disclosing as ‘containing’ the event and leaking it if exposed to the ‘right’ circumstances [21]. Such a unidirectional model falls short when contrasted with the experiences of women subjected to GBV, where inconsistencies, back-and-forth trajectories are common, and where the ‘listener/supporter’ places a crucial role in the process [22,23,24].

Our point of departure instead is that disclosing GBV is a social practice, a co-construction of meaning between the one disclosing and the one(s) receiving/supporting/listening. Disclosing is, thus, not the one-way communication of some ‘inner truth’ but a joint construction and sense-making of events, where the ‘listener/supporter’ plays an active role in the process and where what is discussed and how it is discussed is framed by discourses. From such a perspective, we consider that a discursive approach to disclosure [21] is more fruitful.

What women disclose is shaped by how it is received, it is not a matter of professionals supporting women to articulate their unvarnished truth. Instead, it is about how professionals and women together construct the concept of GBV. It is also important to note that the experiences that are disclosed between professionals and women are also shaped by the available concepts and dominant frames existing in one particular place and time [24].

Disclosing GBV may not necessarily be preceded by becoming aware of the problem, but it may be that both develop in parallel; for example, women may make sense of their experiences and become aware that what they are undergoing is GBV when they talk with someone about it and (jointly) put a name to the experience [24]. Disclosing and concealing GBV can both be used strategically and may be best approached not as mutually exclusive processes but as happening interactively and concurrently [25].

There is research describing disclosure of GBV as a stepwise process and enumerating the factors and contexts that facilitate or hinder this process [6,15,19,20,26]. What is lacking is research analysing the process of GBV disclosure per se, bringing together and contrasting the experiences and perceptions of professionals who (should) support disclosure and of women subjected to GBV (who are expected to disclose it).

## 2. Methodology

### 2.1. Study Setting and Design

This study was carried out in the Madrid region, which includes the capital of Spain and surrounding towns, plus a large rural area. GBV is common is Spain; according to the last macro-survey on GBV conducted in 2019, the life-time prevalence of GBV is 57.3%, and 19.8% of women have suffered GBV during the last year. Prevalence is higher among younger women (life-time prevalence of 71.2% among women ages 16–24) [27]. The ‘Organic Law of Measures of Integrated Protection Against Gender-Based Violence’ approved in 2014 established a number of strategies to prevent and respond to GBV in different sectors (police, law, media, education, health, social services). The relevance of identifying GBV in the different public services that women may contact is stressed in several policies and guidelines [3]. When it comes to young women, our previous research in the Madrid region shows that there remain a considerable number of barriers related to accessibility, acceptability, equity, appropriateness, and effectiveness [28].

The material analysed in this paper comes from a larger study exploring GBV against young women in relation to their access to resources. We conducted 13 semi-structured interviews with young women (aged 16 to 36) who had been subjected to GBV and had been in contact with public services in relation to this. Six of the participants were employed, four unemployed, two were studying, and one was employed and studying. Two of the participants had children. We also interviewed 17 professionals (14 women and 3 men). These professionals worked with GBV in different areas such as psychology, social work, the police, nursing, psychiatry, and social education. Three worked in public institutions that worked at the state level, twelve at the level of the region/municipality, and two at civil society organisations. More information about theoretical sampling and participants’ profiles has been previously published elsewhere [28]. We did not gather data on ethnicity/race from any participants (more under limitations). All the qualitative interviews conducted in the larger study were re-analysed in the current paper but with a different aim in focus.

### 2.2. Data Collection

Face-to-face interviews with professionals were carried out between March and July 2019 by one of the authors (ECT). The interview guide in this case included topics such as their perception of the current situation of GBV among young women, how they worked with addressing GBV, how they perceived those other services as addressing GBV, as well as barriers and facilitators in detecting and providing support for GBV and proposals for improvement. The interviews were recorded and lasted between 45 and 90 min.

In the case of the women, and due to the COVID-19 pandemic, the interviews were carried out digitally (via phone call, video call, or email) between April and September 2020 by ECT. We asked them about their experiences of GBV, how they had identified that they were suffering GBV, how they sought support, and their experiences of this process. Topics related to informal and formal help resources, as well as proposals for improvement in the provision of support, were also covered. More information about the topics discussed in the interviews can be found elsewhere [28].

### 2.3. Data Analysis

We analysed the interviews using reflexive thematic analysis as described by Braun and Clarke [29,30]. After transcribing the interviews, we read them several times to familiarise ourselves with the data. Then we started coding the interviews conducted with professionals, keeping the initial study aim in mind. After the coding, we developed candidate themes and wrote short descriptions of them, which were then further discussed within the team. We then went back to the data to contrast these candidate themes with the code list and transcripts, and during this process the themes were revised and refined. After agreeing within the research team upon a preliminary theme structure, we started coding the transcripts from the young women and developed preliminary themes from that material, which we contrasted with the ones developed from the material extracted from the professionals’ interviews. We then developed an extensive report that combined all the material. It was at this point that we read the literature more carefully, looking for concepts that could help us delve deeper into the analysis. At this point, we chose ‘a discursive approach to disclosure’ as a conceptual frame. With this in mind, we again revised the structure of the themes, went back to the transcripts of both professionals and young women, and refined the structure to the final four themes that we present here. The process was conducted in the original Spanish and the translation of quotes only took place once the final structure of themes had been developed. In the presentation of results, we identify the quotes from women who have been subjected to GBV with the letter W and those from the professionals with a P.

With our discursive approach to disclosure, we see the two data sets (from professionals and young women) as complementary—young women reconstruct their stories of violence within the frames that are available to them at a particular moment in time and also based on their experiences during their meetings with professionals. At the same time, professionals’ understandings of violence are shaped by their encounters with young women who have been subjected to violence. Together, they construct a way to understand the process of disclosure. Finally, we, as researchers, enter into the picture and also co-construct an understanding of the process that is shaped by our pre-understandings, theoretical positions, experiences, etc.

Building upon a discursive approach to disclosure we analyse the interviews from young women and professionals not to provide a description of the specific situation of disclosing GBV in the Madrid region but to theorise around the concept of disclosing GBV. In that sense, our results contribute to a better understanding (and problematisation) of the process of disclosing GBV and open up questions that are relevant for the way public services (in general) address GBV. Consequently, we consider that the questions we open up are relevant beyond the specific setting of Madrid and Spain.

### 2.4. Reflexivity

Berger describes reflexivity as ‘the process of a continual internal dialogue and critical self-evaluation of researcher’s positionality as well as active acknowledgement and explicit recognition that this position may affect the research process and outcome’ [31] (p. 19). Our positions as middle-age, white academic feminists without experience of violence sets a power imbalance with the young participants while placing us closer to the professionals interviewed. We are aware of the power imbalance (especially with the young women interviewed) and agree with Fontes in that, ‘researchers in family violence are more powerful than the participants by virtue of living their own lives in safety (assuming the researchers are not themselves in a violent relationship)’ [32] (p. 55). We remained aware of this imbalance throughout the research process and tried always to avoid a patronising stance both during the interviews and the analysis. Keeping a critical perspective but at the same time not criticising the perspectives of participants can be a tricky balance, but we have tried to do the former without engaging in the latter.

Our different degree of familiarity with the participants was helpful in questioning each other’s preunderstandings in the research team. As first author, I came up with the idea based on my interest on analysing public services’ focus on identifying violence through asking, which was what triggered my curiosity when I first read the transcripts. Some of the other team members conducted the interviews and were closer to the participants’ narratives. This allowed interesting discussions, where my conceptually driven analysis (but also building upon my previous experience of conducting qualitative interviews with women subjected to GBV and professionals) became sometimes questioned by the more inductive analysis from some of the other authors. Through continuous discussions and keeping these different lenses we became more alert to each other’s pre-understandings and allowed to conduct a more nuanced analysis of the material. Peer-debriefing with other researchers also allowed to scrutinise these preunderstandings, but one limitation is that we did not share our preliminary findings with activists, professionals, or women subjected to GBV, which would have illuminated other aspects.

### 2.5. Ethics

Each participant received and signed an informed consent form by email, in which the objectives of the study were explained, as well as the reason for the interview. Their anonymity and the confidentiality of the opinions they expressed were guaranteed. Furthermore, they were assured that their participation was voluntary and that they could withdraw at any time during the interviews if they so wished. The project was approved by the Ethics Committee of the Carlos III Health Institute, protocol CEI PI 61_2019-v3.

## 3. Results and Discussion

From the analysis of the material, and building upon a discursive approach to the disclosure of GBV, we developed four themes about how disclosure is perceived by both the women and the professionals: (i) as a conversation between acquaintances; (ii) as ‘no solution’; (iii) as a possible prerequisite for action; and (iv) as difficult because GBV is normalised.

### 3.1. Disclosure as a Conversation between Acquaintances More Than Unidirectional (Professional) Asking—(Woman) Telling


*I was talking to my friend and I was wearing shorts. He asked me what had happened because I had a bruise, the leg was black. I used to tell this friend many things, then I told him that it had been him [her boyfriend at the time] who did it. And my friend asked me how come I allowed someone to do that to me. And I answered that ok, he’s already asked for forgiveness, and my friend replied that there are some things that shouldn’t be forgiven. It was there and then that I started thinking about it.*
(W9)


*The psychological attention at the centre didn’t help me at all, there was no feeling, I didn’t trust her 100%. It wasn’t because of the psychologist that I realised I was suffering GBV, it was more because of the other women who were in the waiting room of the clinic where we talked to each other and then I started reconsidering things.*
(W10)

As these quotes from women describe, becoming aware of GBV and telling someone about it were not two sequential steps in a unidirectional process, but instead went hand in hand. Becoming aware of violence occurs as part of a conversation (it is a matter of *talking* about GBV instead of *telling* about GBV): more than women first identifying GBV within their lives and then telling someone about it, it was more that through talking with someone about GBV women also started to realise that they were experiencing it. These experiences align better with a discursive approach to disclosure, in which disclosure is not about communicating some inner untouched story but about articulating a story and making sense of it during the process of telling it [21,24]; it is a two-way process and a continuum, not a one-off asking and telling. How the conversation unfolds matters, as the following quotes describe:


*She [one of her teachers] gave me the care and empathy that I needed at that moment. I had been talking with her for four years, she was the one who followed up all my process, and she had a patience and empathy with me that was fierce.*
(W12)


*Empathy, to feel that they are believed, that’s key. You need to be very careful, especially the first time she talks about it […] she needs to feel that she is believed.*
(P1)

W12, one of the women participating, and P1 a social worker in one municipality, both reflected upon the importance of empathy and trust. In line with disclosing as a two-way avenue, how the woman’s story is received affects the identification process. When disclosing GBV, talking cannot be a monologue but requires someone who listens and believes the story, validates it, and is not judgemental. Both professionals and women talked about the importance of women feeling that the person to whom they disclose believes them and does not judge, stands by them and has patience. If, instead of empathy and trust, the woman encounters judgement and disbelief, she may end up doubting whether what she has finally identified as GBV and found the courage to disclose, is actually violence at all.

In general, professionals were not the ones the women talked with: friends and relatives usually came first. This is not surprising, and the literature stresses that, in general, women subjected to violence talk first with relatives and friends rather than with professionals [33]. In our interviews, the women described how they may talk with professionals later on, once they had more clarity and determination about the resources and support they needed. But sometimes disclosure also took place with professionals:


*I went to the psychologist at the 24 h centre and I started telling her everything that came to my mind about what I’ve lived through with him and she [the psychologist] started to put a name to everything. She started telling me that [...] the anxiety I was feeling was normal, that I was not responsible, that what he was doing was illegal, that he was harming my health, and I started seeing everything from that point of view: ‘shit, I’ve survived this, this is serious and I didn’t know it’, and she started showing me a new world, to tell the truth.*
(W12)

Disclosure is achieved through dialogue, and the ‘recipient’ of the information is not neutral. Instead, as the above quote depicts, the psychologist ‘started to put a name to everything’ that the woman disclosed. Or, in the quotes at the beginning of this theme, it was friends who challenged W7 and W9 to reconsider their experiences as violence. These findings problematise the assumption that disclosure is a unidirectional process and that the one who listens is a passive recipient. From a discursive perspective, instead, it is the one listening who shapes what is disclosed in relation to providing names and legitimising or delegitimising the claims of the one disclosing [24]. Existing discourses have effects on the intelligibility of the experiences disclosed—specifically, women may need to ‘learn’ the proper ways to tell their stories of violence in order to be understood by those listening [34,35].

### 3.2. Not Disclosing GBV Is Harmful but Disclosing Is Not the Solution


*I was very anxious, because I’d normalised it so much, and I hadn’t dared to disclose it.*
(W2)


*When this [sexual violence] is kept silenced, it generates issues in the body as well. Fibromyalgia, there are many, […] Not everyone with fibromyalgia has been abused. But there are many women who have been subjected to abuse who have fibromyalgia. And that’s related to this issue about containing, silence, guilt, shame, with not being able to put it into words. In the end, if you don’t put it into words, then your body will somehow, you have to release all that.*
(P15, psychologist working in a municipal centre for women who have been subjected to sexual violence)

Both professionals and women described the harmful consequences of ‘holding things inside’ and not disclosing violence. Professionals described a variety of symptoms that they associated with GBV, and that women may complain about for a while before labelling their situation as GBV. Depression, addictions, anxiety, and fibromyalgia were all considered potential indicators that should make professionals suspect the possibility of GBV before women disclose it. From this perspective, disclosure is framed under a ‘hydraulic model of mental health’; disclosure becomes a strategy to release bad experiences that ‘the psyche accumulates’ and ‘refill it with remedy messages’ [21]. Unnamed violence is represented as ‘leaking’, becoming embodied in signals that should make professionals suspect the situation even before the women themselves are aware of it and/or disclose it. Professionals then become the ones able to ‘read’ the signals of GBV, even before the woman becomes aware of it.

While our results reveal that not disclosing was related to ‘physical ailments’, the act of disclosing was not represented as empowering and therapeutic per se. Other studies have already questioned this dominant assumption that disclosure and/or talking is inherently beneficial. Carbin, for example, highlights how disclosure can become an extra demand, a responsibility, or a requirement for ‘proper victims’ to break the silence [34]. Instead of merely being a way to relieve stress and gain access to services, disclosure can also become a responsibility, a marker that distinguishes good and bad victims, and a requirement for getting support [24,34,35,36,37]. While such effects did not come up very strongly in our material, what became salient was that disclosure did nor mark THE end of all the women’s problems.


*Nowadays well..., I feel like a mess, alone, with a precarious job, in need of a workshop to value myself more.*
(W12)


*I feel stronger, but at the same time I feel fearful on many occasions, and very insecure, I think this insecurity will be with me always, more or less, but always there.*
(W3)

As these quotes exemplify, women experience violence as something with long-term effects on their lives, effects that linger long after the violence has been disclosed. Disclosure can sometimes be useful, empowering and rewarding, and can open up access to shelter and financial and legal support. However, for the women, labelling their experiences as violence and telling someone about it, even telling professionals who can enable the possibility of accessing support, did not mean that all the problems came to an end. Disclosure has the potential to empower women, but it can also be a way to oversimplify and transform complex, confusing experiences into an apparently coherent narrative for the sake of rendering them intelligible, measurable, and suitable for acting upon [34,35]. Placing disclosure at the centre of responses can thus come with false hope and exaggerated expectations, which disregards the fact that, for women subjected to GBV, disclosure does not solve their problems. While disclosure can be a door opener to support, publicising it as ‘the solution’ risks minimising the reality that, after disclosing, there may still be a long way ahead for women. The criticism has also been made that the focus on disclosure risks positioning it as a measure of success, disregarding the fact that success is dependent on what comes after women disclose and that unfortunately what comes after disclosure is not always what women require [4].

### 3.3. Is Disclosure a Prerequisite for Action?


*The way they helped me out was to get me into a women’s shelter. I stayed there longer than a year, and the first year I still couldn’t... I couldn’t consider in my head that I had been subjected to violence. To be honest, I think I still find it hard to process it […] Obviously it has the name of violence and abuse, but […] the name I gave it was: different realities. […] Obviously it’s not normal, and it’s not how I should live, but it’s there, it’s a reality... different, but one. I never got into labelling it, because existing terms were too big for me at that moment, I thought: ‘It can’t be’. You see cases on TV and listen to how people talk, but I never thought that I was going through that. Even at the shelter, I thought ‘fuck! my friends here have suffered violence’, but to say that I myself have suffered violence, even today it takes me a lot to take it on board.*
(W10)

In this quote, W10, one of the women interviewed, described how speaking about her experience and labelling it as violence was still something that was difficult for her. At the time of the interview, she was actually living in a shelter for female victims of GBV, but she still had not fully assimilated that what she had gone through was GBV. She described a long journey during which self-questioning/self-doubting and un-naming were commonplace. The literature has also pointed out that women who have been subjected to GBV may name their experiences as violence at one point, only to back off and doubt that they were victims of GBV at another. Or, women may not recognise their experiences as GBV because they perceive them as different from what media messages describe as GBV and they do not recognise themselves in the stereotypical portrayals of women who have been subjected to GBV [22,23,24,38].

W10’s experience compels us to question the stepwise model that represents self-realisation as a required prior stage before disclosure can take place and disclosure as a prior stage for action—which is mainly described in the literature as leaving the abusive partner [39]. As our findings point out, action may not always be the last step, especially in the case of young women who may have been exposed to or witnessed other forms of violence within the family. Instead, action sometimes precedes both self-realisation and disclosure. We are not the first to question this linearity in the process; as Enander summarises in her research on experiences of female victims of GBV in Sweden: disclosing violence openly may occur after women have already taken some action [22,23].

The literature has also pointed out that disclosing GBV comes with consequences and expectations, and one such expectation is that the woman will leave the abusive relationship. However, this is not an easy decision for many women [24,39]. In turn, professionals may feel frustrated and helpless if women do not leave their abusers after revealing the abuse [40]. Disclosing GBV to professionals and then going back to or continuing with the abusive partner is surrounded by shame, which makes disclosing a big step. If disclosing becomes attached to expectations and responsibilities on the part of the woman disclosing, then it may be easier to fully disclose to others and put a name to the experience, retrospectively, once the woman is in a better and safer place and has decided to act as she is expected to. Instead of disclosure opening up into action, it becomes legitimated by having taken action. However, it is important to notice that studies with young women emphasise that disclosing past experience of GBV can also be perceived as worthless [10].

Disclosure is thus not always a neat and linear process, and women may not always require full disclosure in order to take action. However, at the same time, the lack of self-realisation and disclosure were described as crucial problems that needed to be solved. This was especially salient in the interviews with professionals. For example, P13, a nurse in a primary healthcare centre explained:
*That’s happened to me several times, when women come to me, they’re not suffering GBV, none of them. When I tell them, they say no, no. But when I open their eyes and I tell them look, what you’re telling me is GBV. I mean, that he controls your money in the bank, that’s GBV. That he doesn’t allow you to dress how you like, that’s GBV. Then, when I put what they’re telling me into words, and I tell them, then they realise perfectly well, and then they want to access resources, they want help.*(P13)

P13 highlights that, when she first meets them, women are not aware that their experiences are GBV, or do not acknowledge it. It is the professionals’ role to help them put their experiences into words and label them as such. While one can read the words of P13 as being directive or leading, across the interviews P13 and other professionals stressed that they needed to be respectful of women’s path and ‘tempo’ and be patient, but, at the same time, professionals needed to be one step ahead, able to notice GBV even before the woman herself did, and to guide the conversation in the ‘right’ direction. They have to help women become aware of their situation, because when they manage to do that, women ‘*realise perfectly well, and then they want to access resources, they want help*’. Supporting women to disclose GBV is what opens up the possibilities for change.

Such a perspective aligns with how access to resources works in Spain. It is no longer required that women make a formal legal denunciation to access resources (even though research shows that professionals have an orientation towards denouncing, see [24]), but still a woman subjected to GBV in Spain needs a professional (not just anyone, but those who specifically work with GBV) to confirm this in order for her to access certain resources. Disclosing to a professional and the professional confirming this then becomes a requirement for accessing support.

This centrality of the role played by professionals in supporting disclosure and legitimating women’s claims of GBV can be linked with a reasoning based on expert knowledge [24,35,37]. Thus, expert knowledge is crucial in several ways: in supporting women to disclose, in being able to read the signals even before women themselves become aware or disclose, and in confirming women’s experiences and labelling them as GBV. This produces a shift in who the expert is and where the source of this expertise lies: from women to professionals, from subjective experience to professional knowledge [35,36,37].

Finding language to express women’s experiences of violence was at the core of the women’s liberation movement. As Kelly puts it: ‘A vital part of feminist work around sexual violence has been to provide names that describe women’s experience’ [41] (p. 139). Now that responding to GBV is becoming institutionalised in most countries, and is increasingly a responsibility of the welfare system, providing names for women’s experiences also becomes part of such responses. While this increases public responsibility for the problem of GBV, it also risks shifting the focus from women to professionals, and de-gendering GBV [24,42].

Our findings thus highlight a friction. On the one hand, disclosure is difficult and comes with expectations, which makes it easier for women to disclose after they have taken action. On the other hand, disclosure is a crucial tool for institutional responses that both support women and legitimise expert knowledge on GBV.

### 3.4. The Problem of GBV Being Normalised


*I don’t remember a home without violence. I lived with [my father] beating my mother and my sister. My sister almost died from a brutal beating that left her unconscious. I remember my mother always in bed, crying, unable to get up […] My later romantic relationships built upon this environment. I didn’t have any memories that weren’t related to violence and fear. It wasn’t until five years ago that I became able to distinguish between being well treated and being badly treated. For me, all that was normalised.*
(W2)


*I used to tell her [the psychologist] that it was normal, because my neighbour was going through the same, my cousin was going through the same, my other friend the same, and I thought that was normal for all women.*
(W10)


*They lack awareness, sometimes, that what they are living [is GBV]. Among young women, we’re not talking so much about physical violence; that exists, but to a lesser extent, it’s control, submission, other expressions of violence that are not so obvious to the victim, so she doesn’t ask for help. So, at first, they don’t consider themselves victims, it takes time, and they don’t know about the resources, because they haven’t looked for them, because they’ve never seen themselves as victims, until they get into very extreme situations that make them ask for help.*
(P1, social worker)

From the quotes above, it can be seen that disclosing GBV is experienced as difficult because, for some women, GBV was so normalised that they did not identify it as something worth disclosing. P1, a social worker, argued that there is a problem of a ‘*lack of awareness*’. She also reflected that, when GBV is psychological, it becomes harder to identify and name it as violence and for women to perceive themselves as victims. Both professionals and young women described the continuum of violence (across the life-course and also widespread among most women) and how they perceived that it was difficult to distinguish between what was violence worth disclosing and what was just part of women’s lives. This argument, that women are not aware of GBV because they consider it commonplace and pervasive in their lives, is frequently heard in both the media and research, sometimes as a way to denounce the pervasiveness of violence in women’s lives. Such an argument, however, also risks becoming a victim-blaming technique—placing the responsibility on women for their own situation, portraying them as ignorant and/or complicit [43].

We want to question the assumption that women do not disclose because they have normalised certain forms of violence and that this problem could be solved by asking specific questions as some standardised protocols propose—to ask about specific acts such as punches, hitting, threats, etc. [44]. Instead, we propose that how women understand violence and tell their stories, and how professionals receive those stories, does not take place in a vacuum but is instead shaped by media and legal discourses of what ‘real’ (or punishable) violence is and what it is not and of who is a ‘real victim’ or a ‘real batterer’ and who is not [21,43]. Definitions around GBV continuously change and evolve, and this carries consequences both for how professionals and protocols ‘identify’ violence and for how women themselves understand and frame their experiences [45]. Our findings, showing that certain types of violence are easier for women to recognise and disclose (physical violence versus psychological violence), may reflect not so much a lack of awareness on the part of women about other types of GBV but rather the types of GBV that the system is more likely to accept as legitimate and thus to trust the victim [44].


*Imagine a girl who has attended a workshop about violence in high school, she’s 15—I’m thinking of an actual case I had—and this is the first time she’s heard about it, but during the workshop she starts to realise that this is happening to her. Then it’s more likely that she, maybe after the workshop, or maybe three days later, or five years later... but she will realise that something is happening to her.*
(P15, psychologist)


*With my class, we went to see Pamela Palenciano’s [a feminist artist who presents a performance about her experiences of GBV] monologue, and I had an anxiety attack, very strong, and when I went out I talked with one of my teachers, and she told me that in my relationship I may be suffering violence, and that I should talk to a professional. She recommended a women’s NGO.*
(W13)

Finally, it is important to notice that some people, mainly professionals, considered that, in a context in which GBV is perceived as normal and commonplace in women’s lives, it becomes crucial to create spaces where women can re-signify the ‘normal’, become aware, and put a name to their experiences of GBV. Beyond one-to-one disclosure, P15 and W13 reflected upon the need for collective spaces and/or encounters where women can question this normalisation and reconfigure their experiences as GBV. Feminism and specific movements such as #MeToo were mentioned here, building upon previous evidence [11]. Other spaces that professionals considered important in de-normalising violence were the promotions, consciousness-raising groups and activities that different institutions and organisations were developing with young people. We can interpret this as signalling that responses relying on individual-based solutions of supporting disclosure may not be enough, and that collective spaces are needed to problematise GBV and challenge societal perceptions of such violence. We can also interpret the fact that the interviewed women mentioned the relevance of these spaces far less frequently as a signal that such initiatives are not reaching all women equally and/or are not recognised as being among the triggers for disclosing GBV in all its forms.

### 3.5. Limitations and Strengths

One possible limitation of the study is that the aim started to be developed after the interviews with the professionals were conducted. In that sense, disclosing was not a specific focus of the questions asked. However, it was the preliminary analysis of the material what encouraged us to analyse this aspect, since professionals repeatedly talked about the process of disclosing. This speaks to the relevance of the topic, and lead to including the exploration of the process of disclosing in the interview guides with women.

Another possible limitation is that interviews with women were conducted via Zoom; while that could have hindered the rapport between participants and interviewer, it was the only possible way due to the COVID pandemic.

In relation to diversity, we have reflected upon our positions (see section on reflexivity), and we include two different perspectives: that of professionals and that of women. Our sample includes diversity in relation to socioeconomic characteristics, but we did not gather information on ethnicity/race. This may fail to capture how the experience and practice of asking and being asked can be shaped by such categories. In relation to diversity, a strength is that we are focusing on young women, not in comparison to adult women but as the group in focus on its own right. When ageism sets adult women as ‘the norm’, we consider that centring on other age groups can be considered as contributing towards diversity in GBV research.

## 4. Conclusions

By employing our discursive approach to disclosure, we identified it as a co-construction between women and professionals and how women’s perceptions of what GBV is, and their stories about it, are shaped by their encounters with professionals and others. These encounters are also shaped by broader discourses of what GBV is. Disclosing GBV has effects; even if it is not the solution per se, it can offer professionals an opportunity to support women. Disclosing makes it possible to respond institutionally to GBV through the figure of the expert professional who is alert to signs, knows how to support disclosure and has the power to legitimate women’s claims of GBV.

Disclosing GBV comes with expectations of how women should act. This means that it is easier to disclose after women have taken the expected actions. Finally, while individual responses to women subjected to GBV are necessary, collective spaces to question GBV are crucial to extend disclosure to all forms of GBV, including those that are more normalised and perceived by young women as a normal part of their daily lives.

This study emphasises that it is crucial to ensure that when women do disclose, their experience extends beyond asking and answering, into an empathic conversation where they feel validated. It is also important that disclosure does not become a requirement but rather one possibility and that other ways of addressing GBV beyond disclosure are also explored—for example, approaches such as the women’s malaise, which addresses women’s unspecific ailments from a gender perspective without the requirement to disclose GBV. Disclosure opens up the opportunity to access resources, but it cannot become the marker of success per se, without further follow up of what happens after disclosure. It is also important to further analyse whether there could be other ways of legitimating women’s stories of GBV without the requirement of professionals’ endorsement.

It is urgent to find ways to leverage the responsibility and expectation placed on women who disclose (e.g., to do something, to leave the abusive partner, to file a denunciation). Professionals need to find ways to fully support women in a way that avoids shaming them for ‘failing’ in relation to what is expected of them. This can strengthen existing institutional responses to GBV.

Finally, this study also stresses that, in order to tackle GBV, individualised solutions, such as professionals supporting women to disclose GBV during individual consultations, are not enough. There is a need for spaces that question structural gender inequality and GBV and that allow them to put names to their experiences and re-signify them. Such spaces, provided by both institutions and the feminist movements, need to reach more women and encompass more forms of GBV.

## Data Availability

The data presented in this study are available on request from the corresponding author. The data are not publicly available due to privacy issues.

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
