# Peer review of "Disclosing Gender-Based Violence: A Qualitative Analysis of Professionals’ and Women’s Perspectives through a Discursive Approach"

_ijerph, 2022, doi:10.3390/ijerph192214683_

Round 1
Reviewer 1 Report
This article problematizes discursive assumptions of GBV disclosure as a positive step in attaining adequate support in healing processes. The treatment of the topic is original as it challenges established assumptions that may be countering or delaying healing processes. Findings assist the design of more complex professional and institutional programs. This article emphasizes the agency of women in making sense of their experiences, the definition of GBV, and its impact on their lives. It underscores the importance of dominant discourses, place, manner and content of disclosure taking place in both professional and non-professional relations as factors affecting the outcomes of telling. Gathering data on racial identity, sexual orientation, socioeconomic class and religion should be considered as factors shaping disclosure processes and outcomes. Conclusions are consistent with the evidence and arguments presented. Methodology is adequate to address the main question posed; however, it could benefit from a discussion of the specific sociopolitical context in which the study took place and a reflection on how this study may apply to other social environments. References are relevant, with some recent research. Tables and figures are lacking. They should be provided.Author Response
This article problematizes discursive assumptions of GBV disclosure as a positive step in attaining adequate support in healing processes. The treatment of the topic is original as it challenges established assumptions that may be countering or delaying healing processes. Findings assist the design of more complex professional and institutional programs. This article emphasizes the agency of women in making sense of their experiences, the definition of GBV, and its impact on their lives. It underscores the importance of dominant discourses, place, manner and content of disclosure taking place in both professional and non-professional relations as factors affecting the outcomes of telling. Gathering data on racial identity, sexual orientation, socioeconomic class and religion should be considered as factors shaping disclosure processes and outcomes. Conclusions are consistent with the evidence and arguments presented. Methodology is adequate to address the main question posed; however, it could benefit from a discussion of the specific sociopolitical context in which the study took place and a reflection on how this study may apply to other social environments. References are relevant, with some recent research. Tables and figures are lacking. They should be provided.
Thanks for encouraging comments. We have now added a brief information about the socio-political context in a new subsection under Methodology (Study setting and design). See on pages 5-6 where it reads:
Study setting and design
This study was carried out in the Madrid region, which includes the capital of Spain and surrounding towns, plus a large rural area. GBV is common is Spain; according to the last macro-survey on GBV conducted in 2019, the life-time prevalence of GBV is 57,3%, and 19,8% of women have suffered GBV during the last year. Prevalence is higher among younger women (life-time prevalence of 71,2% among women ages 16-24) [27]. The ‘Organic Law of Measures of Integrated Protection Against Gender-Based Violence’ approved in 2014 established a number of strategies to prevent and respond to GBV in different sectors (police, law, media, education, health, social services). The relevance of identifying GBV in the different public services that women may contact is stressed in several policies and guidelines [3]. When it comes to young women, our previous research in the Madrid region shows that there remain a considerable number of barriers related to accessibility, acceptability, equity, appropriateness and effectiveness [28].
We have also added an explanation of how we judge the transferability of our results. See on page 8 where it now reads:
Building upon a discursive approach to disclosure we analyze the interviews from young women and professionals not to provide a description of the specific situation of disclosing GBV in the Madrid region, but to theorize around the concept of disclosing GBV. In that sense, our results contribute to a better understanding (and problematization) of the process of disclosing GBV and open up questions that are relevant for the way public services (in general) address GBV. Consequently, we consider that the questions we open up are relevant beyond the specific setting of Madrid and Spain.
There are no tables or figures in this manuscript.
Reviewer 2 Report
This is an interesting study, We will provide the authors with some suggestions for improving the manuscript. The comments will follow the structure and order of the manuscript.
On line 96, the authors wrote: …discourses. From such a perspective, we consider that a discursive approach to disclosure (see e.g., MacMartin, 1999[21]) is more fruitful. Please follow the journal's citation guidelines. Throughout the manuscripts, there are several other authors misquoted.
The authors talk about this being a partial study and that the sample was larger. If this is a partial sample of the study, they should justify the reason why they have extracted this partial sample and not present the results of the study with the complete sample.
They should include a sample section and describe in more detail the characteristics of the sample. This is addressed at the beginning of the methodology and when talking about the material analyzed.
In data collection, the interview should specify the structure, questions, areas covered, etc., in addition to the time frame.
In relation to ethical aspects, the authors should specify in the text the name of the ethics committee and the protocol number and date of approval. This should be included in the text, even if it is indicated at the end of the document.
The methodology section should be revised, clearly identifying the sample, techniques and instruments, and procedure.
Regarding ethical aspects, the authors should specify in the text the name of the ethics committee and the protocol number and date of approval. This should be included in the text, even if it is indicated at the end of the document.
I recommend the authors update the bibliography with some more current sources.I recommend you read the following article.
Barragán-Medero, F., & Pérez-Jorge, D. (2020). Combating homophobia, lesbophobia, biphobia, and transphobia: A liberating and subversive educational alternative for desires. Heliyon, 6(10), e05225. https://doi.org/10.1016/j.heliyon.2020.e05225
Overall, it is an interesting study with an interesting research model. Although it has limitations that the authors themselves acknowledge, I believe the work can be considered for publication when the authors address the suggested changes.
Author Response
Reviewer 2
Comments and Suggestions for Authors
This is an interesting study, We will provide the authors with some suggestions for improving the manuscript. The comments will follow the structure and order of the manuscript.
On line 96, the authors wrote: …discourses. From such a perspective, we consider that a discursive approach to disclosure (see e.g., MacMartin, 1999[21]) is more fruitful. Please follow the journal's citation guidelines. Throughout the manuscripts, there are several other authors misquoted.
We have revised this.
The authors talk about this being a partial study and that the sample was larger. If this is a partial sample of the study, they should justify the reason why they have extracted this partial sample and not present the results of the study with the complete sample.
We have tried to clarify this now. This study was part of a larger study, but from the current paper we also analyzed all the interviews conducted in the larger study. What we meant with the acknowledgement that this paper was part of a larger study is that the aim of the current paper was not planned like that from the start. We re-analized the material with this new research question in mind, because in the preliminary analysis the issue of disclosure was salient, and we were interested in digging deeper into it. We have tried to clarify this better in the text now. See on page 6 where we have added:
The material analysed in this paper comes from a larger study exploring GBV against young women in relation to their access to resources. We conducted 13 semi-structured interviews with young women (aged 16 to 36) who had been subjected to GBV and had been in contact with public services in relation to this. Six of the participants were employed, four unemployed, two were studying and one was employed and studying. Two of the participants had children. We also interviewed 17 professionals (14 women and three men). These professionals worked with GBV in different areas such as psychology, social work, the police, nursing, psychiatry and social education. Three worked in public institutions that worked at the state level, 12 at the level of the region/municipality and two at civil society organizations. More information about theoretical sampling and participants’ profiles has been previously published elsewhere [28]. We did not gathered data on ethnicity/racialization from any participants (more under limitations). All the qualitative interviews conducted in the larger study were re-analyzed in the current paper, but with a different aim in focus.
They should include a sample section and describe in more detail the characteristics of the sample. This is addressed at the beginning of the methodology and when talking about the material analyzed.
We have added some additional information on the sample. The sample is described extensively in another published paper from the research team (see Ref. 28), so we consider that readers interested in the details can read it there as well. See on page 6:
We conducted 13 semi-structured interviews with young women (aged 16 to 36) who had been subjected to GBV and had been in contact with public services in relation to this. Six of the participants were employed, four unemployed, two were studying and one was employed and studying. Two of the participants had children. We also interviewed 17 professionals (14 women and three men). These professionals worked with GBV in different areas such as psychology, social work, the police, nursing, psychiatry and social education. Three worked in public institutions that worked at the state level, 12 at the level of the region/municipality and two at civil society organizations. More information about theoretical sampling and participants’ profiles has been previously published elsewhere [28].
In data collection, the interview should specify the structure, questions, areas covered, etc., in addition to the time frame.
We provide a description of the main topics addressed during the interviews and we now direct the readers to a published paper from the research team where there is a detailed description of the interview guides (Ref 28). See on page 7:
Face-to-face interviews with professionals were carried out between March and July 2019 by one of the authors (ECT). The interview guide in this case included topics such as their perception of the current situation of GBV among young women, how they worked with addressing GBV, how they perceived those other services as addressing GBV, as well as barriers and facilitators in detecting and providing support for GBV and proposals for improvement. The interviews were recorded and lasted between 45 and 90 minutes.
In the case of the women, and due to the Covid-19 pandemic, the interviews were carried out digitally (via phone call, video call or email) between April and September 2020 by ECT. We asked them about their experiences of GBV, how they had identified that they were suffering GBV, how they sought support and their experiences of this process. Topics related to informal and formal help resources, as well as proposals for improvement in the provision of support, were also covered. More information about the topics discussed in the interviews can be found elsewhere [28].
We have decided not to add more details in this paper to not make it too lengthy.
In relation to ethical aspects, the authors should specify in the text the name of the ethics committee and the protocol number and date of approval. This should be included in the text, even if it is indicated at the end of the document.
We have added this. See on page 10:
The project was approved by the Ethics Committee of the Carlos III Health Institute, protocol CEI PI 61_2019‐v3.
The methodology section should be revised, clearly identifying the sample, techniques and instruments, and procedure.
We have provided more information about the participants (see answer to previous query). We have also added a section on study setting to further contextualize the study. As explained before, in order to not make this paper longer, and since the details of the sample and interview guides are explained in detailed in another published paper from the team, we have decided not to add more details here.
Regarding ethical aspects, the authors should specify in the text the name of the ethics committee and the protocol number and date of approval. This should be included in the text, even if it is indicated at the end of the document.
See answer to previous comment
I recommend the authors update the bibliography with some more current sources. I recommend you read the following article.
Barragán-Medero, F., & Pérez-Jorge, D. (2020). Combating homophobia, lesbophobia, biphobia, and transphobia: A liberating and subversive educational alternative for desires. Heliyon, 6(10), e05225. https://doi.org/10.1016/j.heliyon.2020.e05225
We thank the reviewer for this suggestion. It is an interesting article, but we do not find it relevant for the topic addressed in our paper.
Overall, it is an interesting study with an interesting research model. Although it has limitations that the authors themselves acknowledge, I believe the work can be considered for publication when the authors address the suggested changes.
Thanks